# Arene-Ruthenium(II) Complexes with Carbothiamidopyrazoles as a Potential Alternative for Antibiotic Resistance in Human

**DOI:** 10.3390/molecules27020468

**Published:** 2022-01-12

**Authors:** Ewelina Namiecińska, Magdalena Grazul, Beata Sadowska, Marzena Więckowska-Szakiel, Paweł Hikisz, Beata Pasternak, Elzbieta Budzisz

**Affiliations:** 1Department of the Chemistry of Cosmetic Raw Materials, Medical University of Lodz, Muszynskiego 1, 90-151 Lodz, Poland; ewelina.namiecinska@umed.lodz.pl; 2Department of Pharmaceutical Microbiology and Microbiological Diagnostics, Medical University of Lodz, Muszynskiego 1, 90-151 Lodz, Poland; magdalena.grazul@umed.lodz.pl; 3Department of Immunology and Infectious Biology, Faculty of Biology and Environmental Protection, University of Lodz, Banacha 12/16, 90-237 Lodz, Poland; beata.sadowska@biol.uni.lodz.pl (B.S.); marzena.wieckowska@biol.uni.lodz.pl (M.W.-S.); 4Department of Molecular Biophysics, Faculty of Biology and Environmental Protection, University of Lodz, Pomorska 141/143, 90-236 Lodz, Poland; pawel.hikisz@biol.uni.lodz.pl; 5Department of Organic Chemistry, Faculty of Chemistry, University of Lodz, Tamka 12, 91-403 Lodz, Poland; beata.pasternak@chemia.uni.lodz.pl

**Keywords:** antimicrobial, antibacterial, antioxidant, arene-ruthenium(II) complexes, antibiotic resistance, antibiotics, carbathioamido pyrazole derivatives, hexafluorophosphate ion

## Abstract

To meet the demand for alternatives to commonly used antibiotics, this paper evaluates the antimicrobial potential of arene-ruthenium(II) complexes and their salts, which may be of value in antibacterial treatment. Their antimicrobial activity (MIC, MBC/MFC) was examined in vitro against *Staphylococcus aureus*, *Staphylococcus epidermidis*, *Enterococcus faecalis*, *Pseudomonas aeruginosa, Proteus vulgaris* and *Candida albicans* and compared with classic antibiotics used as therapeutics. Selected arene-ruthenium(II) complexes were found to have synergistic effects with oxacillin and vancomycin against staphylococci. Their bactericidal effect was found to be associated with cell lysis and the ability to cut microbial DNA. To confirm the safety of the tested arene-ruthenium(II) complexes in vivo, their cytotoxicity was also investigated against normal human foreskin fibroblasts (HFF-1). In addition, the antioxidant and thus pro-health potential of the compounds, i.e., their nonenzymatic antioxidant capacity (NEAC), was determined by two different methods: ferric-TPTZ complex and DPPH assay.

## 1. Introduction

One of the most prevalent threats to patients is posed by antibiotic resistance. In such cases, therapeutic options have run out, and mortality from infections caused by resistant microorganisms is increasing. Almost all clinical strains are becoming resistant to one or more classes of antibiotics due to overuse of antibiotics [1]. The resulting selection pressure favors the persistence and spread of resistant bacteria, including *Streptococcus pneumoniae* strains resistant to penicillin and third generation cephalosporins, *Staphylococcus aureus* resistant to methicillin, and enterococci resistant to high concentrations of aminoglycosides, vancomycin and linezolid [2,3,4]. There is, therefore, a pressing need for a multidirectional response to counter this threat, including an intensive search for new drugs and development of more effective therapeutic strategies [5].

The diversity of available metals, ligands and geometries makes metal complexes a very promising direction for drug development [6]. As metal complexes can adopt three-dimensional structures (most organic molecules possess only one-/two- dimensional topologies), they provide the possibility to design a wide variety of antimicrobials. 

The chemical, physicochemical and biochemical properties of metal complexes may vary based on the properties of their components (central metal, ligands and salts) alone. Their biological activity depends on several features, such as the metal ion and structure of the ligand, or ligands, to which the metal ion is bound. Moreover, their mechanism of action is determined by the size, charge distribution, molecular shape and redox potentials of the complexes [7]. It is noteworthy that metal complexes may influence microorganisms through different modes of action: exchange or release of biologically-active ligands, redox activation and catalytic generation of toxic species (reactive oxygen species, ROS), as well as depletion of essential substrates [8], providing them with the ability to inhibit enzyme activities, disrupt membrane function or damage nucleic acids [9].

Over the past two decades, most studies of organometallic compounds have focused on their anticancer activity. While cisplatin is currently the “golden standard” in this regard, it demonstrates numerous side effects and drug resistance related to its use [10,11,12]. One group of metal complexes that has been frequently investigated for anticancer activity includes complex compounds containing ruthenium ions; for instance, NAMI-A and KP101 are in phase 2 clinical trials [13,14,15,16]. Ruthenium complexes demonstrate a variety of biological properties and are probably best known for their anticancer properties [17,18,19,20,21]. In ligand exchange reactions, labile ruthenium complexes are able to coordinate with nucleic acids, while inert compounds, usually with polypyridyl ligand(s), intercalate with DNA and RNA [22,23]. The characteristic structural geometry of organometallic complexes allows them to interact with protein structures. Additionally, ionic complex compounds are positively charged, which supports their binding not only to DNA and RNA but also to other intracellular targets such as phospholipids and some proteins [6,24,25,26]. Recently, ruthenium compounds have been investigated as potential antimicrobial agents [23,27].

Our previous studies examined the biological properties, such as the biostatic and biocidal activity, of selected areno-ruthenium(II) complex compounds containing Cl^−^ ions with appropriate pyrazole ligands **1a**–**1d** (C-3 -alkyl, C-5 -aryl, -hydroxyl) [17]. The aim of the present study was to continue our research into the antibacterial properties of compounds **2a**–**2d** [17] and their salts containing hexafluorophosphate anions (**3a**–**3d**). The activity against a wide range of microorganisms, expressed as MIC/MBC/MFC, was determined using broth microdilution. To understand the mechanism of action of the most promising compounds, their ability to induce bacterial cell death was evaluated, as well as their influence on DNA, and time-kill experiments were conducted. The study also examines their synergistic antimicrobial effect in combination with commonly used antibiotics (Figure 1) [15]. In addition, the cytotoxicity of the compounds was also investigated against normal human foreskin fibroblasts (HFF-1), as well as their antioxidant properties.

## 2. Results

### 2.1. Characterization of Newly Obtained [PF_6_] Arene-Ruthenium(II) Salts (Complexes **3a**–**3d**)

All the newly-synthesized salts were obtained by an exchange reaction between dichloro *p*-cymenoarenoruthenium(II) containing the corresponding carbathioamidopyrazole ligand (with a methyl or ethyl substituent at the C-3 position and an arenyl and hydroxyl/carbonyl substituent at the C-5 position) [17,28]. The ammonium hexafluorophosphate salt (NH_4_PF_6_) is in this case a Lewis base. The reaction, therefore, results in the precipitation of the corresponding salt (NH_4_^+^Cl^−^) and the formation of a new arene-ruthenium(II) salts with hexafluorophosphate ions **3a**–**3d** (Figure 2).

The new salts **3a**–**3d** were synthesized under an argon atmosphere at a 1:1 molar ratio of complex: salt, with anhydrous dichloromethane used as a solvent (Figure 2). The reactions were carried out at room temperature to obtain new colored solid or powder compounds **3a**–**3d**. For the obtained salts **3a**–**3d** were characterized by spectroscopic methods (^1^H NMR, ^13^C NMR, FT-IR, ESI-MS) and elemental analysis.

#### 2.1.1. NMR and FT-IR Spectra

The ^1^H NMR and ^13^C NMR spectra data recorded in DMSO-d_6_ for all arene-ruthenium(II) salts **3a**–**3d** are given in the Experimental section. Ligands **1a**–**1d** and compounds **2a**–**2d** were prepared according to the procedure described in the literature [17,28]. The chemical shifts and intensities confirmed the proposed structures in the synthesis procedure. The chemical shifts of *p*-cymene produced signals ranging from 2.40 to 3.01 ppm as septet, those of the alkyl substituents ranged from 0.90 to 263 ppm, while those of the protons in the pyrazole ring ranged from 3.13 to 6.80 ppm. The amino group protons produced signals from 9.41 ppm to 11.20 ppm. The ^13^C NMR resonance of the C=S group produced signals for complexes with chloride ions at **2a**–**2d** 145.70, 152.01, 175.88 [17] and 175.18 ppm, whereas protons for the newly-obtained complexes with hexafluorophosphate ions yielded 146.14, 152.01, 175.20 and 175.25 ppm for **3a**–**3d**, respectively.

In FT-IR studies, compounds **3a**–**3d** demonstrated vibrations indicative of amino group-stretching, i.e., in the range 3443 to 3091 cm^−1^. Characteristic shifted regions for pyrazole *v*(C=N) group were observed (1604 to 1647 cm^−1^). The *v*(C=S) measurements revealed stretching vibrations from 824 to 846 cm^−1^, suggesting that the sulfur atom in substituents C=S may be a possible site of ruthenium(II) coordination. Moderate single bands were observed for *v*(C-N) from 1359 cm^−1^ to 1402 cm^−1^.

#### 2.1.2. ESI-MS Spectra

The compounds were analysed by electrospray using a Varian MS 500 spectrometer, which allows perforation of spectra in various modes (negative and positive) and tandem spectra. Electrospray is one of the most sensitive mass spectrometry techniques used to determine the molecular weight of compounds, and to confirm the presence of existing ions in biological solutions. Samples **3a**–**3d** were dissolved in a methanol-water system in a ratio of 95:5. As expected, only the *m*/*z* = 145 Da signal was observed in the ESI-MS spectra of arene-ruthenium(II) complexes **3a**–**3d** in negative ion mode. This peak derives from the [PF_6_]^−^ anion, which we have not seen in previously tested compounds [17]. The ESI-MS results, in both positive and negative ion modes, are shown in Table 1.

The main signal for each compound was the ion [ArRuL]+ at *m*/*z* 390.0 **3a**, 418.1 **3b**, 454.1 **3c**, 392.1 **3d**. Na+ adducts were observed only for **3a**; however, they were less intensive. The next part of the analysis was the study of the main ions by tandem spectrometry. The MS/MS fragmentation of ions containing a chlorine atom leads to ions of the type [ArRuL]^+^. The ESI-MS/MS spectra indicate that the precursor ion at *m*/*z* 392.1 fragmented to yield a product ion at *m*/*z* 333.1 by loss of H_2_N=C=S caused by the cleavage of the *N*-Ru bond (Figure 3 and Appendix A). The ESI -MS/MS/MS spectrum of the product at *m*/*z* 333 indicated further fragmentation into several products (Figure 3).

The ESI-MS/MS spectra presented in Figure 3 and Appendix A show that the precursor ion at *m*/*z* 392.1 was fragmented first to a product ion at *m*/*z* 333.1, and finally to the product at *m*/*z* 231, by the loss of H_2_N=C=S caused by the cleavage of an *N*-Ru bond (Figure 3).

### 2.2. Antimicrobial Activity

Representative pathogens, as well as opportunistic microorganisms and human microbiota, including species from staphylococci, enterococci, *Pseudomonas*, *Proteus* and *Candida* yeasts, were used to evaluate direct antimicrobial activity of arene-ruthenium(II) complexes with carbotioamidopyrazole derivatives **3a**–**3d**. For comparison purposes, the activity of selected antibiotics was also assessed. The minimum inhibitory concentrations (MIC) and minimum bactericidal/fungicidal concentrations (MBC/MFC) of the tested compounds are presented in Table 2.

Similar to the activity of arene-ruthenium(II) complexes **2a**–**2d** described previously [17], only Gram-positive bacteria such as *S. aureus*, *S. epidermidis* and *E. faecalis* were susceptible to complexes **3a**–**3d**. Arene-ruthenium(II) complex **3c**, similar to **2c** (see Namiecinska et al.) [17], demonstrated the strongest biostatic and biocidal activity. Most of the compounds did not inhibit the growth of Gram-negative bacteria such as *P. aeruginosa* and *P. vulgaris* or the yeast *C. albicans* up to 1000 µg/mL. However, complexes **3a** and **3b** demonstrated weak fungistatic activity, with MIC values of 500 and 1000 µg/mL, and **3d** displayed weak bacteriostatic and bactericidal effects against *P. aeruginosa*. It was shown that NH_4_PF_6_—a substrate for the new complexes **3a**–**3d,** as well as a dimer, and most of the ligands used for the synthesis (see our previous publication [17]) remained inactive. For comparison the MIC/MBC/MFC values were also assessed for selected antibiotics (Table 2). *E. faecalis* was found to be very sensitive to Gen, with MIC/MBC below the range of concentrations tested (2–128 µg/mL), while *C. albicans* was resistant to Flu, with MIC/MFC above the range of concentrations tested (0.25–16 µg/mL).

### 2.3. Synergistic Effect of Arene-Ruthenium(II) Complexes (**2a/3a**–**2d/3d**) with Selected Antibiotics

Assuming that arene-ruthenium(II) complexes as potential therapeutics in future could be used alone or together with antibiotics, their synergistic effects with oxacillin (Oxa) and vancomycin (Van) were tested against *S. aureus* ATCC 29213 and *S. epidermidis* ATCC 12228. The MICs of the antibiotics, used alone or in combination with arene-ruthenium(II) complexes, were determined by a broth microdilution assay followed by microbial growth on solid media. Following this, the correlations between both antimicrobials (synergy, additive, indifference, antagonism) were calculated based on the fractional inhibitory concentration (FIC) using the formula FIC = MIC antibiotic in combination/MIC antibiotic alone. The obtained results are presented in Table 3.

All arene-ruthenium(II) complexes revealed synergy with Oxa and almost all with Van against *S. epidermidis*, decreasing the MICs of these antibiotics. However, the most promising were **2b**, **2c** and **3b**, which also expressed synergistic effects against *S. aureus*. Neither dimer nor NH_4_PF_6_ demonstrated such synergistic activity with the tested antibiotics.

### 2.4. Time-Kill Experiments and Cell Lysis

The time-kill test is used to analyze time-dependent or concentration-dependent antimicrobial effects [30]. Time-kill studies were performed to determine whether compounds **2a**–**2c**, **3a**–**3c** were bacteriostatic or bactericidal against *S. aureus* ATCC 29213 and *S. epidermidis* ATCC 12228, as these compounds were the most effective molecules against these strains. Compounds **2a**, **2b**, **3a**, **3b** were shown to be bacteriostatic against *S. aureus* and *S. epidermidis* (Figure 1A and 1B, respectively). A similar effect was demonstrated for the compounds **2c** and **3c** against *S. aureus* (Figure 1A). In all of these cases, the bacteria continued to grow but displayed fewer colony-forming units (CFUs) compared to controls at each time point. In addition, compounds **2c** and **3c** were bactericidal against *S. epidermidis* ATCC 12229 (Figure 1B). However, although they completely inhibited *S. epidermidis* growth during all tested time points, some regrowth was observed at 24 h. It is possible that the tested compounds were not able to completely kill all bacterial cells present (inoculum) at the used concentration, or that the compounds decomposed after longer incubation times.

Antibiotics, such as B-lactams, may cause cell lysis and cell death by inhibiting the synthesis of the bacterial cell wall. Therefore, the next stage of the study examined the potential of the tested compounds to lyse bacterial cells and to determine the mechanism of any such antimicrobial activity. The method examined the effect of the compounds against the bacterial cell wall and cell membrane, which are major targets for many antibacterial agents. Exposition to **2a**, **3a**, **2b** and **3b** used at 1 × MBC was not found to change OD_600_ values of *S. epidermidis* ATCC 12228 cells (data not presented), suggesting that the treated bacterial cells were not lysed. Although compounds **2c** and **3c** demonstrated a bactericidal effect on *S. epidermidis* (Figure 1B), cell lysis studies indicated a quite moderate decrease in OD_600_ (Figure 2). This indicates that only partial cell lysis occurred, but it should be noted that OD measurements provided only an indication of total biomass obscuring the light path, and no insight into viable biomass. Therefore, OD measurements systematically underestimated lysis in late phase of growth, and additional, more advanced, studies of tested compounds, such as cell lysis inductors, are needed.

### 2.5. Fluorescence Microscopy Studies

Nucleic acid dyes are widely used to analyze the physiological state (viability) of individual microbial cells. The simultaneous use of two fluorescent dyes (propidium iodide and Hoechst 33258), with different fluorescence spectra and cell penetration mechanisms, enables the identification of cells at various physiological stages: live, early and late phase of death. Propidium iodide (PI) is negatively charged and penetrates only the cells with damaged cell membranes. It allows the identification of necrotic cells or cells in late stages of death. Hoechst 33258, on the other hand, penetrates freely through the intact membrane enabling identification of living cells and the cells in the early stages of cell death. Both Gram-negative and Gram-positive bacteria can be stained with Hoechst or propidium iodide. The cells in the early stages of death, with impaired integrity of their cell membranes, tend to stain brighter with Hoechst dye than living cells, and show blue fluorescence. In contrast, PI enters the cells in the late stages of death when the membranes are severely damaged.

Following the antimicrobial activity of the analyzed compounds **2a**–**2d** [17] and newly compounds **3a**–**3d** and determined MIC values (Table 2), the viability of *S. aureus* ATCC 29213 and *S. epidermidis* ATCC 12228 following treatment with the tested compounds was determined microscopically. For this purpose, the cells were stained with Hoechst 33258 and PI fluorescent dyes after 24-h incubation with the test compounds used at MIC concentration. Obtained results, shown in Figure 3 and Appendix A, confirmed that arene-ruthenium (II) complexes demonstrate biostatic or biocidal activity. The tested compounds were characterized by various degrees of influence on *S. aureus* and *S. epidermidis* cells, manifested as the changes in their physiological state. The greatest changes in bacterial viability were observed for *S. aureus* and *S. epidermidis* cells treated with compounds **2c** and **3c** (Figure 3). Most of the bacterial cells showed intense blue fluorescence. Moreover, it is worth emphasizing that bacterial cells with red fluorescence also appeared in the field of view after the penetration of propidium iodide, which indicates a high degree of the cell membrane degradation. Less changes in cell survival for both strains were observed after treatment with **2a**, **2b** and **3a**, **3b** (Appendix A). Moreover, no cells were observed in the late stages of death (red fluorescence). In the case of **2d** and **3d**, no significant changes in the viability of *S. aureus* and *S. epidermidis* cells were observed (Appendix A).

### 2.6. Cleavage of pUC57 DNA

Bacterial plasmid DNA usually exists in a supercoiled (SC, I) form. Endonucleases or damaging agents can alter SC states to open-circular (OC, II) or linear (L, III) isoforms. As supercoiled DNA is compacted, it migrates rapidly in agarose gel. Single strand breaks (SSBs) result in the creation of open-circular DNA forms and, due to its larger size, this DNA migrates more slowly in agarose gel. In contrast, double strand breaks (DSBs) result in the formation of linear DNA, which migrates more slowly than SC but more quickly than OC in agarose gel. Therefore, plasmid DNA can be used to identify DNA damage occurring as an effect of metal complex activity [31].

Our findings indicate that compounds **3b** and **3c** are able to cut plasmid DNA (Figure 4). Compound **3a** did not affect DNA structure at a concentration of 100 μM; however, it caused DNA sedimentation at 200 μM and higher, which prevented any analysis. Compound **3c** induced DNA sedimentation at 200 μM. Interestingly, for compound **3b**, no correlation was observed between the concentration of the tested compound and the amount of OC and L forms of DNA.

Results showed that tested ruthenium (II) complexes **3b** and **3c** may nick the DNA Form I into Form II and III in a similar way to complexes **2b** and **2d** studied previously [17], and chemical nucleases.

### 2.7. Cytotoxic Activity

The cytotoxicity of arene-ruthenium(II) complexes **3a**–**3d** was tested against human foreskin fibroblast culture HFF-1 (ATCC-SCRC-1041) as regular eukaryotic cells (Figure 5).

Similar to arene-ruthenium(II) complexes with pyrazole derivatives **2a**–**2c** [17], complexes **3a**–**3c** demonstrated cytotoxicity against regular eukaryotic cells, particularly above concentrations of 250 μM (Figure 5). IC_50_ values of 379.5 μM, 354.6 μM and 264.3 μM were achieved for **3a**, **3b** and **3c**, respectively. However, complex **3d** with an IC_50_ value of 892.0 μM, like complex **2d** [17], did not exhibit significant cytotoxic effect in the range of concentrations tested (3.9–500 μM). In the presence of **3d**, HFF-1 viability did not drop below 75% (Figure 5).

### 2.8. Measurement of Ruthenium Complexes with Chloric **2a**–**2d** and Hexafluorophosphoric **3a**–**3d** Anions’ Nonenzymatic Antioxidant Capacity (NEAC)

To assess the antioxidant activity of the tested ruthenium complex compounds, two in vitro assays were performed, i.e., the free radical DPPH and TPTZ assays. These methods allow fast determination of nonenzymatic antioxidant capacity. The obtained results indicate that all investigated complexes, **2a**–**3d**, are able to interact with stable free radical DPPH and reduce Fe(III) ions. Both methods confirmed that the analysed compounds have antioxidant properties.

Results obtained by the ferric-TPTZ method shown that the tested ruthenium complex compounds had lower antioxidant capacity than the Trolox control (Figure 6). Compounds **2b**, **2a**, **3b**, **3a** had much better antioxidant activity than other tested complexes; all demonstrated particularly high antioxidant activities at higher concentrations (30 µmol/L–60 µmol/L) and their ability to remove free radicals was concentration dependent.

The percentage inhibition of DPPH for these complexes ranged from about 30% to 50%. Compounds **2c**, **2d**, **3c** and **3d** indicated very low antioxidant capacity across the entire range of applied concentrations. The percentage DPPH inhibition of tested compounds ranged from about 10% to 30%, and the highest used concentration (60 µmol/L) corresponded to approximately 12–15 µmol/L Trolox concentration for Fe (III) reduction. The results obtained for the DPPH radical scavenging are shown in Figure 7.

## 3. Discussion

While the biostatic and biocidal activity of chosen arene-ruthenium(II) compounds with pyrazole ligands **2a**–**2b** have been presented previously [17], the *p*-cymene-ruthenium(II) complexes with carbathioamidopyrazole ligands are an intriguing group of compounds with unique properties and require further study. Therefore, the present study was performed to better understand the mechanism of action of compounds containing [Cl]^−^ ions **2a**–**2d** [17] and their new salts with hexafluorophosphate ions **3a**–**3d**. For all newly obtained arene-ruthenium(II) salts **3a**–**3d,** the presence of [PF_6_]^−^ ions was confirmed by ESI-MS.

The biological activity of metal complexes has been found to depend on the type of metal ion and structure of the ligand/ligands to which they are bound. Thus, the size, charge distribution, molecule shape and redox potential of the complexes affect their mechanism of action [7]. Our current studies show that the nature of the ligand structures of *p*-cymene-ruthenium(II) complexes, i.e., containing alkyl (methyl and ethyl), hydroxyl or phenyl groups in the pyrazole ring, affects their antimicrobial activity to varying extents. Many examples have been presented of the relationship between antimicrobial activity and complex compounds with different anionic forms [32,33].

The salts of *p*-cymene-ruthenium(II) complexes with hexafluorophosphate ions presented herein **3a**–**3d** possessed biostatic/biocidal activity. Similar to the *p*-cymene-ruthenium(II) complexes with chloride ions **2a**–**2d**, they appear to target Gram-positive bacteria (*S*. *aureus*, *S. epidermidis* and *E. faecalis*) [17]; this suggests that the effect in the *p*-cymene-ruthenium(II) complexes is not dependent on any exchange of counterions ([PF_6_]^−^ or [Cl]^−^). However, it should be mentioned that salt forms are more stable and/or more soluble for the purposes of drug formulation. Salts containing [PF_6_]^−^ ions are of great interest, as some (e.g., 1-butyl-3-methylimidazolium hexafluorophosphate (C_4_MIM][PF_6_]), 1-hexyl-3-methylimidazolium hexafluorophosphate (C_6_MIM][PF6]), 1-octyl-3-methylimidazolium hexafluorophosphate (C_8_MIM][PF_6_])) have been studied as penicillin V solubility enhancers that can increase antimicrobial activity [34]. It should be noted that the cationic forms in *p*-cymene-ruthenium(II) complexes with carbathioamidopyrazole ligands (**2a**/**3a**–**2d**/**3d**) significantly affect antimicrobial activity. In this study, there were no observed any changes in the spectral characteristics at the peak absorptions for the arene-ruthenium(II) complexes (**2a**/**3a**–**2d**/**3d**) over time (Appendix A). Such a result suggests there are no structural alterations and allows us to conclude that the complexes are stable in aqueous solution. Additionally, the presence of alkyl substituents (methyl in **2a**/**3a** and ethyl group in **2b**/**3b**) in the C-3 and C-5 position of the pyrazole had a significant influence on the antimicrobial activity of tested complexes against selected Gram-positive bacteria strains. A similar effect was observed for the phenyl group in the C-3 position and the hydroxyl group in the C-5 position in the pyrazole ring of **2c**/**3c** complexes.

The effective antibacterial concentrations of arene-ruthenium(II) complexes (**3a**–**3d**) are still much higher than the those of classic antibiotics (Table 2). However, it is important to point out that the possible multidirectional mechanisms of their antimicrobial activity, such as ROS generation, cell lysis and interaction with important biomolecules such as DNA, may delay the development of microbial resistance. Moreover, we anticipate that such metal complexes may be used to support classic antibiotic therapy rather than replace it. Therefore, the study also tested the synergistic effect of arene-ruthenium(II) complexes with oxacillin (Oxa) and vancomycin (Van) against *S. aureus* and *S. epidermidis*. Oxa is an *β*-lactam antibiotic and is active against methicillin-sensitive staphylococci, while Van is a glycopeptide antibiotic recommended for use against methicillin-resistant staphylococci (e.g., MRSA) [35,36]. In our studies, the ruthenium(II) complexes demonstrated a synergistic effect, decreasing the effective concentration of the tested antibiotics. The correlations observed depended mostly on the bacterial strain, but the compounds **2b** and **3b** exerted a synergistic effect in all tested configurations (Table 3). It has been reported that complex formation with some metal ions, e.g., Bi(II) and Cu(II), may increase the antimicrobial activity of fluorochinolone antibiotics such as ciprofloxacin or norfloxacin. It has been proposed that metal ions may influence the solubility of quinolone and its transition across the bacterial cell wall and membrane [37]; however, while this previous work concerns the direct interaction of the antibiotics with metal ions (complexation), whereas the present study examines the synergistic effect of arene-ruthenium(II) complexes with carbothiamidopyrazoles as separate compounds on selected antibiotics.

Similar synergy has been noted against *S. aureus* by the thiadiazole ligand, its Zn(II) complex and kanamycin [38]. In this case, as the MIC of kanamycin combined with both compounds was reduced equally (eight-fold compared to kanamycin acting alone), it was suggested that groups other than the moieties involved in the formation of metal complex were responsible for the interactions with kanamycin [38]. No such conclusion can be drawn in the present study, as both the dichloro(*p*-cymene)ruthenium(II) dimer and NH_4_PF_6_ (which are reagents in complexation) did not show any synergistic effect when used in combination with Oxa or Van.

As noted above, the mechanisms of antimicrobial activity demonstrated by metal complexes are multidirectional. They can catalyze a redox cycle for glutathione oxidation, leading to an increase in reactive oxygen species (ROS) level [39]. It is well documented that antimicrobial compounds may cause cell dysfunction and death by ROS generation and thereby the oxidation of biomolecules, which alters bacterial metabolism and homeostasis [40,41,42]. However, it is also possible that metal complexes may undergo a reversible interaction with important biomolecules, such as DNA, RNA or proteins. In the case of the ruthenium metal center, noncovalent interactions may occur, resulting in a biostatic/biocidal effect caused by target biomolecule dysfunction [39,43].

To investigate potential mechanism of antimicrobial activity of the tested ruthenium(II) complexes, time-kill experiments were performed, i.e., those investigating the ability to induce bacterial cells death, lysis and DNA degradation. Compounds **2a**–**2c** and **3a**–**3c**, were used as theydemonstrated the most favorable MIC results against *S. aureus* and *S. epidermidis*. Time-kill curves are useful for determining the kinetics of bacterial killing and, together with the minimum bactericidal concentration (MBC), may indicate whether an antibacterial agent is bacteriostatic or bactericidal [44]. Information that the tested compound demonstrates bacteriostatic/bactericidal activity in vitro is useful for an initial estimation of the potential antibacterial action of agents. Most of the complexes tested herein appear to have bacteriostatic effects against the tested Gram-positive bacteria. The only exceptions are compounds **2c** and **3c**, considered as bactericidal for *S. epidermidis*; in addition, compounds **2c** and **3c** were also found to be able to inhibit *S. aureus* cell growth more efficiently than the other tested complexes. Interestingly, the *S. aureus* cells demonstrated intensive cell regrowth following incubation for more than 6-8 h with compounds **3a**, **3b**, **2a** and **2b**.

Bactericidal compounds usually lead to microbial cell death by cell lysis or DNA fragmentation, while bacteriostatic antimicrobials most commonly lead to inhibition of bacterial protein synthesis pathways [45]. For instance,β-lactams bind to transpeptidase PBPs (penicillin binding proteins) to inhibit cell wall synthesis, which may lead to cell lysis [46,47]. It is recommended that severely ill and immunosuppressed patients with bacterial infections receive bactericidal agents such as aminoglycosides, *β*-lactams, fluoroquinolones, glycopeptides, lipopeptides, nitroimidazoles and nitrofurans [48,49,50]. Our present studies with fluorescent dyes confirm that compounds **2c** and **3c** are able to reduce the viability of *S. aureus* and *S. epidermidis* cells via degradation of cell membranes. This process may be connected with the cell lysis caused by these compounds (Figure 2).

Antimicrobial agents (including antibiotics) demonstrate a variety of mechanisms of action against pathogenic microorganisms in vivo and in vitro. Sometimes antibacterial agents that are known to be generally bactericidal may only exhibit bacteriostatic activity in vitro, e.g., linezolid in vitro has bacteriostatic activity against staphylococci and enterococci, but bactericidal activity against streptococci [51]. It must be noted that despite their different modes of action, all bactericidal antibiotics induce similar processes that may lead to cell death via increased cell permeability [52], accelerated respiration [53,54], changes in iron and redox metabolism [55] and ROS formation [22,56], as well as oxidative damage to DNA [57], RNA, and proteins [54]. An in-depth understanding of the modes of action of antimicrobial compounds may help to optimize treatment time and prevent of evolution of bacterial resistance during treatment [58,59].

Other groups of antimicrobials are able to influence bacterial DNA, which may occur by several distinct mechanisms [22]. Some compounds inhibit the synthesis of bacterial DNA by influencing certain replication stages. Antimicrobials may directly induce DNA damage by interacting with their target protein. For example, antibiotics such as rifampicin, quinolones and fluoroquinolones block the DNA gyrase enzyme (topoisomerase II or IV) leaving the nuclease domains intact, resulting in DNA being cut without religation [60,61]. Some antimicrobial agents, e.g., bleomycin, induce double-strand breaks [59]. Many pyrazole-fused heterocyclic derivatives exert DNA cleavage abilities [62]. Imidazophenanthrolines also exhibit biological activity related to DNA binding/cleavage properties [63]. Additionally major aminoglycosides, e.g., neomycin B, can cleave DNA at basic sites [64]. Many metal complexes of Cu(II), Ru(II), Pt(II), Co(II) and Zn(II) exhibit anticancer, antibacterial and antifungal properties via intercalative interactions with DNA [65]. It has been proved that ruthenium complexes are able to form coordinate bonds with nucleic acids [23]. Pyrazole-based transition metal complexes with thiosemicarbazide arms often demonstrate good antibacterial activity against *Escherichia coli* via intercalation to DNA [66]. In the present study, complexes **2b**, **2c**, **3b** and **3c** were found to cut DNA, thus leading to cell death.

As the clinical value of these compounds is closely correlated with their cytotoxicity against normal eukaryotic cells, the present study also examined the cytotoxic effect of tested arene-ruthenium(II) complexes on the human fibroblast HFF-1 line (Figure 5). The results indicate that all our tested complexes can be used safely as antibacterials throughout the entire range of effective concentrations indicated (Table 2). Compounds **3a** and **3b** expressed antibacterial activity in the concentration range of 62.5–125 µg/mL, which corresponded to 109–219 µM solutions for **3a** and 104–209 µM solutions for **3b**. While they demonstrated cytotoxicity against the HFF-1 human fibroblast line above 250 µM, with an IC_50_ above 350 µM (Figure 5). The MIC/MBC of **3c** ranged from 31.2 µg/mL or even less, up to 250 µg/mL (Table 2), corresponding with 49.5 µM and 397 µM solution, respectively. Since the IC_50_ of **3c** against HFF-1 cells reached 264.3 µM (Figure 5), this complex might be used safety up to a concentration of 125 µg/mL. Finally, compound **3d** was the least cytotoxic among all the complexes tested, but also the least active as an antibacterial agent. Therefore, despite having a high IC_50_ against HFF-1 of 892 µM, only compounds with lower effective MIC/MBC (below 500 µg/mL, corresponding to 873 µM) would be safe.

Arene-ruthenium(II) complexes (**2a**–**2d**) have been found to be cytotoxic against three cancer cell lines: HL-60, NALM-6 and WM-115. The best results were observed for compound **2d** against the WM-115 melanoma cell line, which demonstrated an activity more than twice that of cisplatin (IC_50_ values: 7.9 µM versus 18.2 µM respectively) [17]. Compound **2c** indicated good cytotoxicity (IC_50_ values: 26.66 µM) against this cell line. Malignant wounds or surgical sites are known to be easily infected by the microbiota of skin and mucous membranes, such as *S. aureus*, *S. epidermidis* or *E. faecalis* [67,68]. Thus, local application of tested arene-ruthenium(II) complexes could not only reduce the development of melanoma cells, but also prevent such complications as wound infections.

Compounds with antioxidant properties have a positive influence on human health. They demonstrate activity against bacterial inflammations and collaborate synergistically with many antibacterial agents against resistant bacterial strains [69]. Metal complexes with antioxidant properties protect cells and whole organisms from the negative influence of oxidative stress or scavenge free radicals. There are suggestions that the antioxidant activity demonstrated by antibacterial compounds might reduce their toxicity against normal eukaryotic cells, thus decreasing their side effects [70,71]. As the oxidative stress associated with antibacterial treatment could lead to the creation of resistant bacterial strains [72], molecules that act by combining antibacterial and antioxidant activities are promising candidates for new antibacterial agents. As such dual roles appear to be played by compounds with two pharmacophores, they may well be more effective for use in patients with chronic diseases (e.g., cystic fibrosis), who also suffer from persistent colonization by drug-resistant pathogens [73,74]. All tested compounds indicated antioxidant activity: **2a**, **2b**, **3a**, **3b** were all able to scavenge DPPH free radicals and reduce Fe (III). Such free radical neutralization activity may be an important advantage for the use of these compounds as antibiotics.

## 4. Materials and Methods

### 4.1. General Materials

Complexes **2a**–**2d** were prepared as described previously [17]. Ammonium hexafluorophosphate and anhydrous solvents: dichloromethane, isopropyl alcohol, diethyl ether were purchased from Sigma-Aldrich (St. Louis, MO, USA) and Avantor Performance. All are commercially available. The new complexes were synthesized under argon atmosphere conditions. For compounds **3a**–**3d**, the ^1^H NMR and ^13^C NMR spectra were registered in DMSO-d^6^ using Bruker Avance III 600 MHz spectrometer (Bruker, Billerica, MA, USA), where chemical shifts (δ) are given in ppm, and coupling constants (J) in Hz. The IR spectra were recorded in KBr on an FTIR-8400S Shimadzu Spectrophotometer (Thermo Nicolet, Waltham, MA, USA). In turn, the ESI-MS spectra were measured using a Varian 500-MS LC Ion Trap (Varian, Palo Alto, CA, USA), and elemental analysis was performed using a Vario Micro Cube by Elemental analyzer (Langenselbold, Germany) (two experiments were conducted in the Faculty of Chemistry at the University of Lodz). Melting points of the new compounds were determined in open capillary tubes on a Büchii B-540 apparatus (Büchi, Flawil, Switzerland) and were uncorrected. The new arene-ruthenium(II) salts (**3a**–**3d**) were dried according to standard methods. Antimicrobial activity analysis was carried out on selected bacteria and fungi strains in the Department of Immunology and Infectious Biology. The antioxidant activity of the compounds was tested in the Department of Molecular Biophysics at the University of Lodz. The other biological activities described in this article were analyzed in the Department of Pharmaceutical Microbiology and Microbiological Diagnostics at Medical University of Lodz. UV-vis spectra were recorded on a UV 1800 Shimadzu spectrophotometer (Kyoto, Japan) at room temperature.

### 4.2. Methods for Chemical Research

#### 4.2.1. General Procedures for New Arene-Ruthenium(II) Salts **3a**–**3d**

Solutions of appropriate arene-ruthenium (II) complexes **2a**–**2d** [17] (0.9–0.11 mmol) in 3–6 mL of anhydrous dichloromethane were dropped slowly into a solution of ammonium hexafluorophosphate salt (0.09–0.15 mmol) in 3–4 mL of anhydrous dichloromethane. The mixtures were stirred under argon atmosphere at room temperature for four hours. The mixtures were then concentrated under reduced pressure and diethyl ether was added. The precipitates were filtered off as orange **3a**–**3b**, red **3c** and brown **3d** solids.

#### 4.2.2. Spectroscopic Characterization of Salts of the Arene-Ruthenium(II) Complexes (**3a**–**3d**)


**Salt of [(η^6^-*p*-cymene) Ru(1-[amino(thioxo) methyl]-3,5-dimethyl-1*H*-pyrazole)PF_6_ (3a)**


Yield for **3a**: (55.67 mg, 88.66%), orange crystals, mp. 167.1–168.0 °C. FTIR (KBr cm^−1^): *v* (N-H) 3443; *v* (C=N) 1622; *v* (C=C) 1477; *v* (C-N) 1359; *v* (C-H) 1160; *v* (N-N) 1049, *v* (C=S) 846. Anal. Calcd for [C_16_H_24_ClN_3_SRu]^+^ [PF_6_]^−^ (M = 570.87 g/mol): C 33.43 (33.25), H 3.84 (3.67), N 7.40 (7.27), S 5.66 (5.55). ^1^H NMR (600 MHz, DMSO-d^6^) δ (ppm): 1.52, 1.65 (2d, ^3^J_HH_ = 12.0 Hz, 6H, CH(CH_3_)_2_
*_p_*_-cymene_), 2.63 (s, 6H, CH_3_; *_p_*_-cymene_, CH_3 pyrazole_), 3.01 (septet, ^3^J_HH_ = 6.0 Hz, 1H, CH(CH_3_)_2_; *_p_*_-cymene_), 3.13 (s, 1H, CH _pyrazole_), 6.22 (s, 1H, ArCH), 6.57–6.74 (m, Ar CH), 7.22 (s, 1H, CH _pyrazole_), 9.41 (s, 1H, -NH), 11.85 (s, 1H, -SH _pyrazole_). ^13^C NMR (600 MHz, DMSO-d^6^) δ (ppm): 14.21, 17.47, 19.38, 21.96, 23.93 (5CH_3_), 31.00 (CH*_p_*_-cymene_), 81.31, 83.90, 85.70, 86.41, 86.79, 88.39(6CH_Ar *p*-cymene_), 103.90,105.49 (2C_pyrazole_), 115.20 (C_pyrazole_), 146.14 (C=S). ESI MS (*m*/*z*): 421.2(33%), 425.9(51%), 424.9(36%), 428.0(52%), 424.0(22%) [M+1]+; MS(-)ESI (*m*/*z*): 145.2[PF_6_]^−^. UV-VIS. λmax (nm): 283.601.


**Salt of [(η^6^-*p*-cymene)Ru(1-[amino(thioxo) methyl]-3,5-diethyl-1*H*-pyrazole)PF_6_ (3b)**


Yield for **3b**: (49.81 mg, 82.1%), orange crystals, mp. 147.9–148.6 °C. FTIR (KBr cm^−1^): *v* (N-H) 3440; *v* (C=N) 1625; *v* (C=C) 1480; *v* (C-N) 1372; *v* (C-H) 1055; *v* (N-N) 1027, *v* (C=S) 840. Anal. Calcd for [C_18_H_27_ClN_3_SRu]^+^ [PF_6_]^−^ (M = 598.93g/mol): C 36.28 (36.09), H 4.74 (4.54), N 7.03 (7.02), S 5.39 (5.35). ^1^H NMR (600 MHz, DMSO-d^6^) δ (ppm): 1.01, 1.15 (2d, ^3^J_H,H_ = 6.0 Hz, 6H, CH(CH_3_)_2_), 1.22, 1.33 (2t, ^3^J_H,H_ = 18.0 Hz, 4H, CH_2_CH_3_), 2.13 (s, 3H, CH_3_; *_p_*_-cymene_), 2.52 (septet, ^3^J_H,H_ = 6.0 Hz, 1H, CH(CH_3_)_2_), 5.68, 5.80, 5.83, 5.99 (4d, ^3^J_H,H_ = 6.0 Hz, 4Ar CH), 6.80 (s, 1H, CH _pyrazole_), 9.00, 11.20 (2s, 1H, -NH_pyrazole_). ^13^C NMR (600 MHz, DMSO-d^6^) δ (ppm): 11.85, 12.99, 18.69, 21.23, 21.51, 21.85, 23.02 (7CH_3_), 24.06 (CH*_p_*_-cymene_), 81.05, 83.25, 86.08, 86.87, 87.13, 87.98 (6CH_Ar *p*-cymene_), 100.41, 106.19 (2C_pyrazole_), 112.95 (C_pyrazole_), 152.01 (C=S). MS ESI (*m*/*z*): 451.0(21%) 452.0(21%) 452.9(28%) 454.0(44%) 456.1(39%) [M+1]+; MS(-)ESI (*m*/*z*): 145.2[PF_6_]^−^. UV-VIS. λmax (nm): 277.814.


**Salt of [(η^6^-*p*-cymene)Ru(1-[amino(thioxo)methyl]-5-hydroxy-3-phenyl-1*H*-pyrazole)PF_6_ (3c)**


Yield for **3c**: (47.81 mg, 81.40%), red crystals, mp. 188.2–189.4 °C. FTIR (KBr cm^−1^): *v* (OH) 3189; *v* (N-H) 3091; *v* (C=N) 1647; *v* (C=O) 1587; *v* (C=C) 1454; *v* (C-N) 1391; *v* (N-N) 1087, *v* (C=S) 824. Anal. Calcd for C_20_H_22_ClN_3_0SRu]^+^[PF_6_]^−^ *H_2_O (M = 652.50 g/mol) C 42.55(42.33), H 4.27(4.02), N 8.32(8.59), S 4.76(4.91). ^1^H NMR (600 MHz, DMSO-d^6^) δ (ppm): 0.90, 0.97 (2d, ^3^J_H,H_ = 6.0 Hz, 6H, CH(CH_3_)_2_), 1.96 (s, 3H, CH_3*p*-cymene_), 2.40 (septet, ^3^J_H,H_ = 6.0 Hz, 1H, CH(CH_3_)_2_), 4.69, 4.75, 5.09, 5.18 (4d, ^3^J_H,H_ =6.0 Hz, 4H, Ar), 5.56 (s, 1H, CH _pyrazole_), 7.62 (s, 3H, Ar CH), 7.97 (s, 2H, Ar CH), 10.33 (s, 1H, -OH_pyrazole_), 10.80 (s, 1H, -NH_pyrazole_). ^13^C NMR (600 MHz, DMSO-d^6^) δ (ppm): 19.78, 21.37, 22.36 (3CH_3_), 30.71 (CH_p-cymene_), 80.34, 82.62, 85.51, 86.20, 87.66, 88.16, 100.63, 128.78, 130.06 (12CH_Ar_), 135.01 (C_pyrazole_), 165.84 (C=O), 175.20 (C=S). MS ESI (*m*/*z*): 488.0(2%) 489(2%) 490.0(3.1%) 492.0(2.5%) [M+1]+; MS(-)ESI (*m*/*z*): 145.2[PF_6_]^−^. UV-VIS. λmax (nm): 279.100.


**Salt of [(η**
**
^6^
**
**-*p*-cymene)Ru(1-[amino(thioxo)methyl]-5-hydroxy-3-methyl-1*H*-pyrazole)[PF_6_] (3d)**


Yield for **3d**: (40.91 mg, 66.17%), dark brown crystals, mp. 155.6–157.2 °C. FTIR (KBr cm^−1^): *v* (NH) 3252; *v* (OH) 3128; *v* (C=O) 1771; *v* (C=N) 1604; *v* (C=C) 1474; *v* (C-N) 1402, *v* (N-N) 1024; *v* (C=S) 840. Anal. Calcd for C_15_H_26_ClN_3_OSRu]^+^ [PF_6_]^−^ (M = 562.41 g/mol) C 33.80(33.21), H 4.35(4.83), N 8.14(7.75), S 5.80(5.91). ^1^H NMR (600 MHz, DMSO-d^6^) δ (ppm): 1.01, 1.12 (2d,bs, ^3^J_H,H_ = 6.0 Hz, 6H, CH(CH_3_)_2_), 2.10 (s, 3H, CH_3*p*-cymene_), 2.39 (s, 3H, CH_3pyrazole_), 2.52 (septet, ^3^J_H,H_ = 6.0 Hz, 1H, CH(CH_3_)_2 p-cymene_), 4.86 (s, 1H, CH _pyrazole_), 5.79 (s, 1H, CHAr) 5.80-5.94 (m, 3H Ar CH), 10.10 (s, 1H, NH), 10.57 (s, 1H, NH). ^13^C NMR (600 MHz, DMSO-d^6^) δ (ppm): 15.65, 17.66, 18.75, 21.46 (4CH_3_), 23.17 (CH*_p_*_-cymene_), 30.81 (CH_pyrazole_), 80.24, 82.76, 84.84, 85.98, 86.83, 87.66 (6CH_Ar_), 103.23 (C_pyrazole_), 165.93 (C=O), 175.25 (C=S). MS ESI (*m*/*z*): 429.1(3%) 434.1(3%) 430.1(4%) 432.1(5%) 431.1(2.5%)[M+1]+; MS(-)ESI (*m*/*z*): 145.2[PF_6_]^−^. UV-VIS. λmax (nm): 282.100.

#### 4.2.3. Physico-Chemical Properties (Stability of Complexes **2a**–**2d** and **3a**–**3d**)

The complexes **2a**–**2d** and **3a**–**3d** were tested for their stabilities in water/DMSO solutions (0.1% DMSO) by UV-vis spectra. The spectra were recorded in the range of 200–800 nm over a time course of 0, 24, 48 and 72 h for **3a**–**3d** complexes and 0, 24 and 48 h for **2a**–**2d** complexes, and compared to each other (see Appendix A).

### 4.3. Materials and Methods for Biological Research (Microorganisms and Culture Conditions)

Reference strains of bacteria and fungi: *Staphylococcus aureus* ATCC 29213, *Staphylococcus epidermidis* ATCC 12228, *Enterococcus faecalis* ATCC 29212, *Pseudomonas aeruginosa* ATCC 25619, *Proteus vulgaris* ATCC 8427, and *Candida albicans* ATCC 10231 (Manassas, VA, USA) were used to screen the antimicrobial activity of the arene-ruthenium(II) complexes **3a**–**3d**, NH_4_PF_6_ and selected antibiotics. The microorganisms were grown for 24 h (bacteria) or 48 h (fungi) in 35–37 °C in tryptic-soy agar (TSA; BTL Sp. z o.o., Łódź, Poland) or Sabouraud’s dextrose agar (SDA; BTL Sp. z o.o., Łódź, Poland) according to nutritional requirements. Microbial suspensions at a density of about 5 × 10^5^ CFU/mL were prepared in Mueller-Hinton broth (MHB; BTL Sp. z o.o., Łódź, Poland) or RPMI-1640 medium (Sigma, St. Louis, MO, USA) for bacteria and fungi, respectively.

### 4.4. Minimum Inhibitory/Bactericidal/Fungicidal Concentration (MIC, MBC/MFC)

The broth microdilution method (as recommended by EUCAST [75]) was used to determine the MIC of arene-ruthenium(II) complexes with carbotioamidopyrazole derivatives **3a**–**3d**, NH_4_PF_6_ and selected antibiotics: oxacillin (Oxa), gentamicin (Gen), and fluconazole (Flu) (all antibiotics were purchased from Sigma, St. Louis, MO, USA). Stock solutions of the lyophilized compounds were freshly prepared in 100% DMSO (POCh, Gliwice, Poland) for the complexes, or in water for injection (Sigma, St. Louis, MO, USA) in the case of antibiotics. Following this, two-fold dilutions were prepared in liquid culture medium to the following final concentration range: 31.2–1000 µg/mL in the first series (the complexes and NH_4_PF_6_ against all tested microorganisms), 1.95–250 µg/mL in the second series (the complexes against staphylococci), 0.03–2 µg/mL (Gen against *P. vulgaris*), and 0.25–16 µg/mL (Flu against *C. albicans*). MIC and MBC/MFC were defined as the lowest concentrations of the compounds inhibiting bacterial/fungal growth or able to kill added microbial inoculum, respectively, and were evaluated as described by Namiecinska et al. [17]. Experiments were carried out in duplicate.

### 4.5. Synergy of Arene-Ruthenium(II) Complexes (**2a**/**3a**–**2d**/**3d**) with Selected Antibiotics

The synergy between arene-ruthenium(II) complexes and the antibiotics oxacillin (Oxa; POL-AURA, Dywity, Poland) and vancomycin (Van; Sigma, St. Louis, MO, USA) against *S. aureus* ATCC 29213 and *S. epidermidis* ATCC 12228 was evaluated by partial checkerboard analysis using broth microdilution. Solutions of Oxa and Van at a final concentration range of 0.06–4 µg/mL against *S. aureus* ATCC 29213 and 0.015–4 µg/mL against *S. epidermidis* ATCC 12228 were mixed with the complexes **2a**–**2d** and **3a**–**3d** used at ½ MIC or dimer and 250 µg/mL NH_4_PF_6_ (all prepared in Mueller-Hinton Broth—MHB; BTL, Lodz, Poland) in 100 µL volumes in 96-well culture microplates (Nunc, Roskilde, Denmark). The microbial suspensions (2–8 × 10^5^ CFU/mL, 100 μL) were then added. In addition, the following controls were prepared: bacteria in MHB alone and in MHB containing 2.5% DMSO as positive controls of bacterial growth, while bacteria were grown in Oxa or Van alone (without the complexes) to check the MIC of antibiotics. After 24 h of coincubation at 37 °C, the MICs of antibiotics alone, or antibiotics in combination, were determined, defined as the lowest concentration able to inhibit visible bacterial growth compared to the appropriate positive growth controls. The inhibition of bacterial growth was confirmed by absorbance measurement (λ = 600 nm) using a Victor2 multifunctional plate reader (Wallac, Turku, Finland) and staphylococcal growth on Tryptic-soy agar (TSA; BTL Sp. z o.o., Łódź, Poland).

### 4.6. Time-Kill Test (Time-Kill Curve)

The time-kill experiments were performed as described previously [76,77]. Briefly, inocula were prepared in TSB medium (TSB; BTL Sp. z o.o., Łódź, Poland) with bacterial suspension of 5 × 10^5^ CFU/mL and were exposed to tested compounds **2a**, **2b**, **2c**, **3a**, **3b**, **3c** at a concentration of 1 × MIC in Mueller-Hinton broth (MHB; BTL Sp. z o.o., Łódź, Poland). All samples were incubated at 35 °C and 80 rpm for varied time intervals (2, 4, 6, 12, 18, 24, 48h). The percentage of dead cells was calculated according to the growth control by determining the number of living cells (CFU/mL) of each tube using the agar plate count method. Samples were immediately serially diluted ten-fold in 0.01 M phosphate buffered saline (PBS, Bimed Lublin S.A., Lublin, Poland) at pH 7.0 and 10 μL aliquots were spot-inoculated in duplicate onto MHA plates. Agar plates were incubated at 37 °C overnight before determining the viable count. Two independent experiments were carried out [76,78].

### 4.7. Fluorescence Microscopy Imaging

*Staphylococcus aureus* ATCC 29213 and *Staphylococcus epidermidis* ATCC 12228 strains were treated with tested compounds **2a**–**2d** and **3a**–**3d** at a concentration 1 × MIC for 24h, while untreated cells were used as a growth control. After incubation, the cells were harvested and a suspension of 1 × 10^5^ cells/mL in PBS was prepared. Following this, 1 µl of the fluorescent dyes Hoechst 33258 and propidium iodide (15 µg/mL each) (Sigma, St. Louis, MO, USA) were added. The obtained mixture was incubated in the dark at room temperature for 15–20 min. After incubation, the cell suspensions, placed on glass slides, were analyzed under an Olympus IX 70 fluorescence microscope (Shinjuku, Tokyo, Japan), using a 360–370 nm UV filter. Photos were taken at 10 × 20 magnification.

### 4.8. Cell Lysis

Bacteriolysis assays were performed as described previously [76]. An overnight culture of *S. epidermidis* ATCC 12229 was used to prepare inocula by diluting to 0.5 McFarland in Mueller-Hinton broth (MHB; BTL Sp. z o.o., Łódź, Poland). Bacterial suspensions were treated with 1 × MBC for all tested compounds, while untreated cells were used as a growth control. All samples were incubated at 37 °C with shaking at 80 rpm for varied time intervals: 0 (control), 5, 10, 15, 20 and 25h. The optical density (OD) of each sample was measured at 600 nm using a Jenway 737501 Genova Nano Micro-Spectrophotometer (Staffordshire, UK) to determine cell lysis. The OD of all samples was determined against the appropriate blank at the appropriate wavelength. Each experiment was performed in triplicate and mean values were determined.

### 4.9. Cleavage of pUC57 DNA

DNA supercoiling plays a key role in gene expression and genome organization [79]. The agarose gel electrophoresis assay was used to check whether the tested complexes **3a**–**3c** are able to cleave DNA. Supercoiled plasmid pUC57 DNA (4634 bp); Thermo Scientific, ABO Sp. z o.o., Gdansk, Poland), was incubated with tested compounds (100, 200 and 300 µM). Next, loading buffer (1 μL) containing 25% bromophenol blue, 0.25% xylene cyanol and 30% glycerol was added to allow the samples to be loaded into 1% *w*/*v* agarose gel. Electrophoresis was performed at 75 V in Tris-acetate- EDTA (TAE) buffer. The gel was stained using Midori green advance DNA stain (Genetics, ABO Sp. z o.o., Gdansk, Poland), and visualised under UV light with the BioRad Gel Doc 2000 system using LABWORK software (Hercules, California, USA). All experiments were carried out in triplicate under the same conditions.

### 4.10. Cytotoxic Activity

The cytotoxicity of arene-ruthenium(II) complexes **3a**–**3d** was determined using the MTT (3-(4,5-dimethylthiazol-2-yl)-2,5-diphenyltetrazolium bromide; Sigma, St. Louis, MO, USA) reduction assay and a human foreskin fibroblast HFF-1 culture (ATCC-SCRC-1041) (LGC Standards Sp. z o.o., Poland) at a final concentration range of 3.9–500 µM. The HFF-1 cells were cultured in Dulbecco’s Modified Eagle’s Medium (DMEM; Sigma, St. Louis, MO, USA) with 15% fetal bovine serum (FBS; BioWest, Nuaillé, France) and penicillin/streptomycin (100 × stock solution; BioWest, Nuaillé, France), 1 × 10^6^ cell/mL (100 µL/well) in 96-well tissue culture plates (Nunc, Roskilde, Denmark) for 24 h at 37 °C/5% CO_2_. The cells were then exposed to arene-ruthenium(II) complexes for 24 h at 37 °C. The control (untreated) cells were exposed to culture medium alone or culture medium containing 2% DMSO to exclude the effect of the solvent used to prepare the stock solutions of the tested complexes. Next, fresh culture medium (100 µL/well) and MTT (1.5 mg/mL, 50 µL/well) were added, and cells were incubated for two hours under the above conditions. MTT was added to blue formazan crystals produced by the metabolically active cells. Finally, the MTT solution was removed and replaced with 100 µL/well of 20% sodium dodecyl sulfate (SDS; Sigma, St. Louis, MO, USA) for 24 h (room temperature) to dissolve the blue formazan. Based on the absorbance (λ = 550 nm) of tested and control cells (100% viability), the percentage of cell viability after exposure was calculated. For each sample, the experiment was performed twice.

### 4.11. Measurement of the Nonenzymatic Antioxidant Capacity of the Ruthenium Complexes

Two methods were used to measure nonenzymatic antioxidant capacity of the analyzed ruthenium complexes—ferric-TPTZ complex and the DPPH assay. The antioxidant activity of ruthenium complex compounds was determined by the DPPH method.

The DPPH assay (1,1-diphenyl-2- picrylhydrazyl; Sigma, St. Louis, MO, USA) is frequently used to evaluate the antioxidant properties of biological compounds in vitro. DPPH is a stable free radical with an unpaired electron on its valence shell. It creates a stable cation radical, and the DPPH solution has a dark purple color with a maximum absorbance in ethanol solution at a wavelength of 517 nm. When encountering an antioxidant, DPPH is reduced, and the purple color disappears. The decrease in absorbance is proportional to the amount of oxidized DPPH remaining in solution and hence the concentration of antioxidants [80]. The ruthenium complexes analyzed were added to the DPPH solution (150 µmol/L). The final concentrations of the samples in the solution were 2, 6, 10, 15, 20, 30, 40, 50, 60 µmol/L. The samples were incubated for 30 min at room temperature. Following this, the absorbance was measured at 517 nm on a microplate reader (BioTek). The percentage of DPPH radical scavenger was calculated using the equation:(1)$$\%\;\mathrm{scavenging}/\mathrm{reduction}=\frac{A_{0}-{\;A}_{1}\;}{A_{0}}\times 100$$
where A_0_ is the absorbance of the control reaction and A_1_ is the absorbance in the presence of the sample or standards. The DPPH radical scavenging activity of Trolox (Sigma, St. Louis, MO, USA) was also assayed for comparison.

The total antioxidant potential of the analyzed ruthenium complexes was determined using the ferric-TPTZ complex (FRAP—Ferric ion reducing Antioxidant Parameter) method. This spectrofluorimetric method is based on determination of the ability of antioxidants to reduce Fe (III) ions. Briefly, TPTZ (2,4,6-iron complex-tripirydylo-S-triazine; Sigma, St. Louis, MO, USA) is reduced under the action of an antioxidant to form an intense blue Fe (II) compound with an absorption maximum at 593 nm [81]. The investigated compounds were mixed with a working reagent consisting of 300 mM acetate buffer (pH 3.6), 10 mmol/L TPTZ in 40 mmol/L HCl and 20 mM FeCl3 freshly prepared in a volume ratio of 10:1:1, and incubated at room temperature for 30 min. The final concentrations of the samples in the solution were 2, 6, 10, 15, 20, 30, 40, 50, 60 µmol/L. At the end of incubation, the absorbance at 593 nm was measured on a microplate reader (BioTek).

Trolox solution was used as a reference compound, with antioxidant activity being expressed as Trolox equivalents calculated on the basis of a calibration curve (obtained data were expressed in µmol/L).

## 5. Conclusions

There is a need for effective alternatives to existing antibiotics. Of the tested p-cymene-ruthenium(II) complexes, the presence of alkyl substituents (methyl in **2a**/**3a** and ethyl groups in **2b**/**3b**) in the C-3 and C-5 position of the carbathioamido pyrazole increased antimicrobial activity against selected Gram-positive bacteria strains. Similar effects were observed for the phenyl group in the C-3 position of the pyrazole ring and the hydroxyl group C-5 position of complexes **2c**/**3c**. It appears that anion exchange [Cl]^−^ and [PF_6_]^−^ does not matter for this effect. Even so, such anion exchange may result in the production of salts with better solubility.

Compound **2b** and its salt with [PF_6_]^−^ decreased the effective concentration of oxacilline against staphylococci. As some of the tested complex compounds indicated bacteriostatic activity, further studies are needed on their influence on bacterial protein synthesis pathways. In addition, compounds **2c**, **3c** were found to have bactericidal potential in the time-kill experiments and were able to cut DNA structure. Therefore, we intend to analyze their influence on bacterial cell structure and DNA in future studies.

Four of the tested compounds, **2a**, **2b**, **3a** and **3b**, demonstrated antioxidant activity based on DPPH free radical scavenging and Fe(III) reduction. Such free radical neutralizing activity may be an important consideration in their potential use as antibiotics.

The combined free radical scavenging and bactericidal activities of ruthenium complexes may have a beneficial effect on reducing the inflammatory status associated with ROS activity. We believe that arene-ruthenium(II) complexes and their salts presented herein offer great potential in the treatment of bacterial diseases, particularly among people affected by cancer or among patients after surgeries dealing with wounds that are difficult to heal.

## Data Availability

Data is contained within the article.

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
