# Peer review of "Arene-Ruthenium(II) Complexes with Carbothiamidopyrazoles as a Potential Alternative for Antibiotic Resistance in Human"

_molecules, 2022, doi:10.3390/molecules27020468_

Round 1
Reviewer 1 Report
The manuscript “Arene-ruthenium(II) complexes with carbothiamidopyrazoles as potential alternative for antibiotic resistance in human” deals with the investigation of biological activity of positive charged ruthenium complexes. This is a very interesting research in dynamically increase area of medical chemistry. The complexes with hexafluorophosphate anions were synthesized from previously described chloride precursors.
In general presented complexes with [PF6]- anion demonstrate similar characteristics as chloride analogs. Contrary, substituents at 3 or/and 5 positions of pyrazole ligand affect increase antimicrobial activity.
The text is written in a good understandable style, the results obtained are beyond doubt.
But several little points I want to clear. The authors do not provide data on stability of complexes and their solubility in water. It is not clear the behavior of substances which synthesized in anhydrous conditions under argon atmosphere in vivo, i.e. aqueous conditions. Also for what reason the authors synthesized complexes under argon but melting points were measured “in open capillary” (line 581)?
A couple of errors are also observed:
- Ratio of intensity of methyl groups of pyrazole “3.13 (s, 1H, CH3 pyrazole), 4.04 (s, 5H, CH3 pyrazole)” (line 607) is stange.
- Line 621: intensity of signal is absent “6.80 (s, CH pyrazole)”.
- “Dry anhydrous” (line 594) is “salt is salty”. I think one of the word is unnecessary.
Author Response
Response to the Reviewer 1
Reviewer 1
Comment 1: The authors do not provide data on stability of complexes and their solubility in water. It is not clear the behavior of substances which synthesized in anhydrous conditions under argon atmosphere in vivo, i.e. aqueous conditions. Also for what reason the authors synthesized complexes under argon but melting points were measured “in open capillary” (line 581)?
Response 1: All synthesis were performed under argon due to the known sensitivity of Ru(II) ion to air oxidation [1]. In the synthesis ruthenium(II) complexes usage argon conditions is a standard techniques [2]. The obtained products show stability in solid (as shown study ESI-MS technique). Moreover, we added spectroscopy UV-Vis studies of the stability of the water solution (0,1%DMSO/water) (please see Supplementary files1). The solubility of the described complexes 2a-d and 3a-d is dependent on the polarity of the solvent. These compounds dissolve best in polar systems such as methanol, ethanol, water, acetonitrile rather than in acetone or hexane.
Literature:
[1] Roberta Signini, Selma E. Mazzetto, Synthesis and Characterization of New Ammine Ru(II) Complexes Containing P(m-tol)3 , P(p-CH3 ) 3 and P(OC5 H11) 3 Roberta Signini a and Selma E. Mazzetto. J. Braz. Chem. Soc., Vol. 13, No. 5, 647-652, 2002
[2] Angel Ruben Higuera-Padillaa, Alzir Azevedo Batistab, Legna Colina-Vegasb, Wilmer Villarrealb and Luiz Alberto Colnago; Synthesis of the [(η6-p-cymene)Ru(dppb)Cl]PF6 complex and catalytic activity in the transfer hydrogenation of ketones. J of Coord Chem.70:20, 3541-3551.
Comment 2: A couple of errors are also observed:
- Ratio of intensity of methyl groups of pyrazole “3.13 (s, 1H, CH3 pyrazole), 4.04 (s, 5H, CH3 pyrazole)” (line 607) is stange.
- Line 621: intensity of signal is absent “6.80 (s, CH pyrazole)”.
- “Dry anhydrous” (line 594) is “salt is salty”. I think one of the word is unnecessary.
Response 2: Thank you for these comments, it were indeed a mistakes in the formulation of the sentence. These sentences have been corrected.

Reviewer 2 Report
The manuscript with reference number “molecules-1527759” reports the synthesis and characterization of four new arene-Ru(II) complexes with carbothiamidopyrazoles ligands. Structural studies were made by means of ESI-MS, FT-IR and NMR spectroscopy. On the other hand, the paper is mainly concerned with evaluating the antimicrobial properties of these compounds and others previously synthesized. For this purpose, authors evaluate the antimicrobial activity in vitro against six microorganisms. The analysis of their ability to induce bacterial cell death, the influence on the DNA and time-killing experiments were made also. Likewise, the synergistic antimicrobial effect in combination with used antibiotics were determined Finally, their cytotoxicity against normal human foreskin fibroblasts (HFF-1) was also investigated and the antioxidant properties were tested.
The novelty of the new compounds synthesized is low since they are the same complex cations as those already published by the authors in 17 (RSC Adv. 2019, 9, 38629-38645). A higher antimicrobial activity than those is not to be expected since the active part of the compounds is exactly the same. And this happens, being low in both cases. However, the work goes in depth on the antimicrobial activity, on their synergistic capacity and on their mode of action. In addition, a quite efficient comparison is made with the compounds already published, including these in the tests performed. The discussion section is quite rigorous. Thus, I consider that the work is interesting and provides interesting information.
However, I have some questions I would like the authors to clarify and there are some aspects that require a minor revision:
- An X-ray diffraction study of the new compounds that would allow us to compare whether there has been a change in the crystalline structures of cation complexes has not been presented. Why?
- The NMR, FT-IR and ESI-MS spectra of the new compounds should be included in the supplementary material as they prove to be very useful information for other researchers.
- Section 2.2.2 and Table 1. I think it would be interesting to compare in this section the results obtained in ESI-MS spectra for 3a-3d with those reported for 2a-2d in [17]. Why is the most abundant fragment [ArRuL] for 3a and [ArRuLCl]+ for 2a? Is there a reason for not detecting the sandwich ions in 3a-3d?
- Section 2.4. It is not clear to me what is the concentration of the compounds in the experiments on synergistic effects. Is the concentration of the commercial antibiotic always the same and the Ru complex is varied? Or is it the other way around? For example, in section 4.5 it is indicated that the concentration of Oxa and Van varies between 0.06-4 µg/mL and Table 3 shows MIC values for Oxa in combination of 0.015 µg/mL, which is lower than the range indicated above. Why? I believe that this section should be presented in a clearer way for the readers.
- Section 2.5. If time-kill experiments are performed with 1xMIC concentrations of 2a-3d, how can bacterial growth be detected for many compounds?
- In Figure 4, not all experiments performed by gel electrophoresis, according to section 4.9, have been represented. Why? I believe that those not shown should be included in the supplementary material.
- Section 4, page 13, line 3. I think the authors should define the abbreviations C4MIM and C6MIM and C8MIM to make it easier for readers to understand.
Finally, minor errors were detected:
- A) Page 3, line 95: The numbering of new section is wrong. It should be “2.1” instead of “2.2”. This mistake about numbering repeats in other pages in lines 110, 129,157,189, 241, 285, 346 and 366.
- B) Page 9, line 299, authors should indicate that MIC values of 2a-2d are presented in ref 17 and not in Table 2 of this work. The sentence now is confused.
- C) Page 11, line 341, “5- DNA +1 µg/mL”, this concentration is not indicated in section 4.9. Is really 100 µg/mL?
- D) Page 12, line 377, “Compounds 2b, 2a, 3b, 3a…” instead of “Compounds 2b, 2a, 3b, 3c…”
These are all reasons why my recommendation is minor revision of the article as currently presented.
Author Response
Response to the Reviewer 2
Reviewer 2 :
Comment 1: An X-ray diffraction study of the new compounds that would allow us to compare whether there has been a change in the crystalline structures of cation complexes has not been presented. Why?
Response 1: We made attempts to determine the grown crystals for hexafluorophosphorus salts, but failed to select a suitable monocrystal. We continue to work on determining the X-ray structure of these complexes and we hope to publish their structures in near future.
Comment 2: The NMR, FT-IR and ESI-MS spectra of the new compounds should be included in the supplementary material as they prove to be very useful information for other researchers.
Response 2: We have included the NMR, FT-IR and ESI-MS spectra in the supplementary materials.
Comment 3: Section 2.2.2 and Table 1. I think it would be interesting to compare in this section the results obtained in ESI-MS spectra for 3a-3d with those reported for 2a-2d in [17]. Why is the most abundant fragment [ArRuL] for 3a and [ArRuLCl]+ for 2a? Is there a reason for not detecting the sandwich ions in 3a-3d?
Response 3:
The results obtained for compounds 3a-d were different from results for compounds 2a-d only by negative ions. As we did not observed significant differences, we did not compare spectra directly. The observation of negative ions was important to us in context of article topic and ESI-MS confirme them.
Regarding the sandwich systems for the remaining compounds, they were present in trace amounts (> 5%) or did not appear, therefore it was unnecessary to include them. The main signals for compounds 2a-d and 3a-d are ions containing only one molecule of the whole complex compound.
Comment 4: Section 2.4. It is not clear to me what is the concentration of the compounds in the experiments on synergistic effects. Is the concentration of the commercial antibiotic always the same and the Ru complex is varied? Or is it the other way around? For example, in section 4.5 it is indicated that the concentration of Oxa and Van varies between 0.06-4 µg/mL and Table 3 shows MIC values for Oxa in combination of 0.015 µg/mL, which is lower than the range indicated above. Why? I believe that this section should be presented in a clearer way for the readers.
Response 4: Thank you very much for your right comment. The concentration of the Ru complex were always the same (1/2 MIC), while the commercial antibiotics were used at varied concentration range. The range of Oxa and Van concentrations tested was 0.06-4 µg/mL against S. aureus ATCC 29213 and 0.015-4 µg/mL against S. epidermidis ATCC 12228, which was not adequately described in Materials and Methods. This important information has been included in the revised text. Moreover, the description of Table 3 has also been improved in the revised text to clear the concentration of the compounds used in the experiments.
Comment 5: Section 2.5. If time-kill experiments are performed with 1xMIC concentrations of 2a-3d, how can bacterial growth be detected for many compounds?
Response 5: In time kill kinetic studies usage of different multiple MIC concentrations of tested compounds is a standard procedure [3-5]. It is possible to see bacterial growth at some point as the experiments are conducted for 24h, while MIC values are calculated after 48-72h of incubation.
Literature:
[3] Paulina L. Páez, Claudia M. Bazán, María E. Bongiovanni, Judith Toneatto, Inés Albesa, María C. Becerra and Gerardo A. Argüello Oxidative Stress and Antimicrobial Activity of Chromium(III) and Ruthenium(II) Complexes on Staphylococcus aureus and Escherichia coli. Biomed Res Int. 2013; 2013: 906912.
[4] Sunniva Foerster, Magnus Unemo, Lucy J. Hathaway, Nicola Low & Christian L. Althaus;Time-kill curve analysis and pharmacodynamic modelling for in vitro evaluation of antimicrobials against Neisseria gonorrhoeae. BMC Microbiology volume 16, Article number: 216 (2016).
[5] Theresa Appiah, Yaw Duah Boakye, and Christian Agyare; Antimicrobial Activities and Time-Kill Kinetics of Extracts of Selected Ghanaian Mushrooms. Evid Based Complement Alternat Med. 2017; 2017: 4534350.
Comment 6: In Figure 4, not all experiments performed by gel electrophoresis, according to section 4.9, have been represented. Why? I believe that those not shown should be included in the supplementary material.
Response 6: We did not see any differences in obtained results whether plasmid DNA was incubated with 50 µM or 100 µM and 300 µM or 400 µM of tested compound, therefore to simplify the graphical presentation of results, we decided to present only photos of gels where DNA was treated with 100, 200, or 300 µM (for compound 2b). However to eliminate the potential misunderstanding, we decided to change the verse from: ” Supercoiled plasmid pUC57 DNA (0.25 lg; Thermo Scientific, ABO Sp. z o.o., Gdansk, Poland), was incubated with tested compounds (50, 100, 200, 300 and 400 µM)” to “Supercoiled plasmid pUC57 DNA (0.25 lg; Thermo Scientific, ABO Sp. z o.o., Gdansk, Poland), was incubated with tested compounds (100, 200 and 300 µM)”.
Comment 7: Section 4, page 13, line 3. I think the authors should define the abbreviations C4MIM and C6MIM and C8MIM to make it easier for readers to understand.
Response 7:
We have define the abbreviations C4MIM, C6MIM and C8MIM in the text.
See below:
C4MIM- 1-butyl-3-methylimidazolium
C6MIM- 1-hexyl-3-methylimidazolium
C8MIM- 1-octyl-3-methylimidazolium
Comment 8: Finally, minor errors were detected:
- A) Page 3, line 95: The numbering of new section is wrong. It should be “2.1” instead of “2.2”. This mistake about numbering repeats in other pages in lines 110, 129,157,189, 241, 285, 346 and 366.
- B) Page 9, line 299, authors should indicate that MIC values of 2a-2d are presented in ref 17 and not in Table 2 of this work. The sentence now is confused.
- C) Page 11, line 341, “5- DNA +1 µg/mL”, this concentration is not indicated in section 4.9. Is really 100 µg/mL?
- D) Page 12, line 377, “Compounds 2b, 2a, 3b, 3a…” instead of “Compounds 2b, 2a, 3b, 3c…”
Response 8: Thank you for these comments.
A)The numbering of section have been rephrased.
B)We have corrected the sentence in line 299.
C)We have corrected this mistake in line 341.
D)We have changed this numbering of compound.
